# The Needs of Adolescents and Young Adults with Chronic Illness: Results of a Quality Improvement Survey

**DOI:** 10.3390/children9040500

**Published:** 2022-04-02

**Authors:** Taryn Allen, Stephanie Reda, Staci Martin, Phoebe Long, Alexis Franklin, Sima Zadeh Bedoya, Lori Wiener, Pamela L. Wolters

**Affiliations:** Pediatric Oncology Branch, National Cancer Institute, Bethesda, MD 20892, USA; taryn.allen@nih.gov (T.A.); sreda@gwmail.gwu.edu (S.R.); martins@mail.nih.gov (S.M.); phoebe.long@nih.gov (P.L.); arf152@miami.edu (A.F.); zadehsl@mail.nih.gov (S.Z.B.); wienerl@mail.nih.gov (L.W.)

**Keywords:** adolescents and young adults, healthcare needs, chronic illness, AYA transition

## Abstract

Adolescent and young adults (AYAs) with chronic illnesses cope with complex issues that require unique psychological support and healthcare services to reduce psychosocial difficulties, improve disease management, and facilitate positive transitions to adult care. Engaging patients and caregivers can help providers understand the specific needs of this population and identify the perceived areas of support. The purpose of this quality improvement initiative is to assess the needs of AYAs with chronic medical conditions at a large government research hospital. Eighty-nine AYA patients (age = 23.5 years; range 13–34) with neurofibromatosis type 1, cancer, primary immunodeficiencies, or sickle cell disease, and a sample of caregivers (*n* = 37, age = 52 years; range: 41–65), completed an anonymized survey that assessed their preferences for a wide range of informational and service-related needs. The results indicate an overwhelming desire for information about general health and wellbeing and disease-specific medical knowledge. The most endorsed item was the need for more information about an individual’s medical condition (72%), which was a primary concern across disease, racial, and gender groups. Demographic and disease-specific needs were also identified. Thus, providing information to AYA patients and caregivers is a critical and largely unmet component of care, which requires the development and implementation of targeted educational and psychosocial interventions.

## 1. Introduction

Large numbers of adolescents and young adults (AYAs) live with chronic medical conditions [1]. The vast majority (90%) of AYAs with chronic illness will survive into older adulthood [2,3] and this percentage will continue to increase, given the continued medical advances and improvements in survival rates [4,5]. Unfortunately, relatively less attention has been focused on AYAs with chronic health conditions, compared to children and adults; thus, the needs of patients within this age band may be less well understood and, as a result, less well addressed by the broader healthcare system.

Among the AYAs with medical conditions, the developmental period is marked by unique physical and psychosocial experiences that co-occur with chronic health needs. Under healthy circumstances, there is a normative increase in social and school or work responsibilities, and AYAs begin to differentiate themselves from caregivers and move closer to their peer group [6]. In addition, adolescence and young adulthood are associated with identity development and substantial physical change. Many of these developmental experiences can be impacted by a chronic illness. For example, on a social level, AYAs with chronic health conditions may need to rely on parents/caregivers to help manage disease morbidities, provide financial support, or handle logistical/pragmatic challenges [7,8], which constrains their level of independence. Further, disease morbidities (e.g., pain and disfigurement), adverse treatment side effects, physician visits, and/or inpatient hospitalizations impede typical social endeavors, such as extracurricular activities, routine socializing, or dating [9,10]. In addition, the developing adolescent brain [11], as well as disease- and treatment-related cognitive impairments, may limit the ability of AYAs to function independently and manage their complex healthcare needs [12]. The disruption caused by chronic health challenges during the AYA period directly impacts developmental processes and creates vulnerability into adulthood [13].

Given the challenges faced by AYAs with chronic illnesses, it is not surprising that research reports lower levels of socioemotional health compared to same-aged individuals without chronic health needs. Poorer social-emotional functioning is evident across AYA populations, including high rates of anxiety and depression in youths with sickle cell disease (SCD) [14] and increased feelings of self-consciousness in youths with lupus [15]. In addition, AYAs with neurofibromatosis type 1 (NF1) have higher rates of emotional and behavioral problems, as well as a diminished quality of life, compared to the AYAs of the general population [16]. Taken together, these examples suggest that AYAs with chronic illness face significant burdens, which make it that much more important that this population receive targeted support during the transitional period.

### Current Research on the Unmet Needs of the AYAs with Chronic Illness

There is a clear need to provide tailored support and services to AYAs with chronic health conditions. Unfortunately, designated support and services for this population are often unavailable [17,18]. Unmet healthcare needs among AYAs with chronic medical conditions are associated with reduced quality of life [19] and put them at risk of negative behavioral health outcomes, such as nonadherence to medication, missed medical appointments, and poor health outcomes [20,21,22]. However, quality of life may be improved both during treatment and after discharge with the addition of specific, tailored informative processes and social services [23]. These services aid the transition from being an AYA who is fully dependent on others for their medical care to an independent adult who manages their own healthcare needs.

Specifically, studies suggest that AYAs want wide-ranging information and guidance from their providers on major life concerns, ranging from brain fog to fertility to finances [19,24]. An equally important unmet need is social support. Studies show that these patients would like to make social connections early on in their care and that many of them report loneliness and disconnection from the AYA community [24,25]. Further, the provision of developmentally appropriate information and activities that enhance social connections within care systems are important needs that remain unmet [26]. Despite the identification of these problems and potential solutions, gaps in AYA services persist.

To best plan for the growing AYA population at our institution, we engaged patients and caregivers in the process to determine the programmatic services that are needed for this age group at the National Institutes of Health through an anonymous survey. Gathering stakeholder input early on is critical for understanding the relevant concerns and will aid in decision-making about the development and implementation of programs and services. Thus, the first aim of this quality assurance study was to examine the needs and services of AYAs with chronic illnesses (i.e., cancer, SCD, NF1, and primary immunodeficiencies) and their caregivers. The secondary aims were to explore the differences between AYAs with different diagnoses as well as among AYAs of different demographic groups, including age, sex, and race. Moreover, this paper aims to highlight the key areas for improving transitional care and services and provides recommendations for those who treat AYAs with chronic illnesses.

## 2. Materials and Methods

### 2.1. Survey Development

A quality improvement and needs assessment survey was developed to evaluate the gaps in services for AYA patients (ages 13 to 34 years) and their caregivers (of pediatric and young adult patients) at a large clinical research hospital in the United States. We solicited expert input from psychologists, social workers, and physicians in the development of the surveys to iteratively generate the domains to assess, determine the response format, and draft the items. The survey spanned topic areas typically encompassed by the construct of health-related quality of life (HRQOL; e.g., physical health, emotional wellbeing, school/work, and social wellbeing), alongside practical healthcare concerns. In addition, the surveys were developed to be brief (approximately 5 min) and were administered via paper and pencil or electronically on an iPad.

The final surveys contained 43 items, and asked patients and caregivers about a broad range of informational needs (i.e., a desire to have more information or knowledge in a specific domain) and service needs (i.e., the desire to have access to specific services of interest). The informational needs section included the following categories: (1) physical/medical health, (2) emotional wellbeing, (3) social and interpersonal wellbeing, (4) school and/or work, (5) general adjustment issues, (6) and practical/legal issues. The service needs section inquired about services of interest, including (1) counseling, (2) recreational services, and (3) social groups or events hosted or organized by the hospital. Slightly different surveys were developed for adolescents versus young adults (e.g., slight wording changes, such as a reference to school *or work* for young adults). Caregivers were asked to rate their own informational and service-related needs as well as their thoughts about the needs of their child. Respondents also were asked to provide basic demographic and medical data (e.g., whole year age, primary medical condition, race, and gender), but no personally identifiable information (PII) was collected. Please see the Appendix A for a copy of the patient and caregiver surveys. For the purposes of this paper, only patient-reported needs and caregiver perception of patient needs are reported.

### 2.2. Procedures

The survey and methods were sent to the Office of Human Subjects Research Protections (OHSRP) at the National Institutes of Health (NIH) to determine whether the IRB review was necessary. Because the survey was deemed a quality improvement/needs assessment survey and did not collect PII, this project was granted an IRB exemption.

Patients between the ages of 13 and 34 years who attended a visit to the National Institutes of Health (NIH) between 2017 and 2019 and had a diagnosis of a chronic or relapse/refractory cancer, SCD, a primary immunodeficiency, or NF1, and/or their caregivers were eligible to participate in this survey. The NIH is a large clinical research hospital, in which patients participate in natural history studies, clinical trials (mostly phases 1 and II), and brief clinical assessments, among other types of studies. Patients who enroll in research at the NIH, typically maintain a home-based physician/hospital with whom they seek regular clinical care for their health condition.

A convenience sampling method was used for this needs assessment. A member of the clinical staff known to them (e.g., a patient’s nurse, social worker, psychologist, and physician) approached the patients and caregivers in the clinic about completing the survey. The staff member provided patients and caregivers with a written description of the project, and verbal consent/assent by the patient and/or caregiver was granted prior to administering the survey. The survey took approximately five minutes to complete and consisted of a list of needs and services that could be endorsed, if it was an area of interest or need (a blank response was coded as a no). Patients and caregivers could be approached separately, so that not all caregivers had a child who participated and vice versa.

### 2.3. Statistical Methods

All statistical analyses were performed using SPSS version 24. Descriptive statistics were used to examine the demographic and medical characteristics of the sample. In addition, the frequency of positive endorsements (i.e., a “yes” response) to each item on the survey was obtained. Pearson’s chi-squared tests were used to examine differences between the groups based on medical condition, demographic variables, including race, gender, and age, and between patients and caregivers. If an omnibus test was significant for medical condition or race, the data was partitioned [27], so that a series of 2 × 2 chi-squared tests could further evaluate the specific groups that diverged in their item response frequency.

In the context of medical condition, the following post hoc comparisons were made: Cancer by SCD; cancer by NF1; cancer by primary immunodeficiencies; SCD by NF1; SCD by primary immunodeficiencies; and NF1 by primary immunodeficiencies. In the context of racial categories, the following post hoc comparisons were made: Caucasian by Black; Caucasian by Asian; Caucasian by multirace; Black by Asian; Black by multirace; Asian by multirace. Of note: in addition to the Caucasian, Black, Asian and multirace participants, there were three individuals who identified as Native American or Alaskan Native in our sample. A decision was made to exclude this group from racial group comparisons, due to the nonrepresentative nature of such a small subsample. In addition, five individuals did not disclose a race and were also excluded from chi-squared tests involving race. Because the purpose of this project is to ascertain patient needs and inform program development better, we did not correct for multiple post hoc comparisons as would be indicated for hypothesis-driven research.

## 3. Results

### 3.1. Descriptive Statistics

In total, 89 patients and 37 caregivers completed the survey. Two individuals who were approached declined to complete the survey; they opted not to provide a specific reason for non-participation. The mean patient age was 23.49 years (SD = 5.9; range 13–34 years), with the overwhelming majority of patient respondents being young adults (≥18 years old; *n* = 75; 84.3%). The majority of patients were male (60.2%) and less than half were Caucasian (41.6%). The primary medical diagnosis of patients surveyed included NF1 (36.9%), cancer (22.6%), SCD (22.5%) and a primary immunodeficiency (16.7%).

Among the caregivers, the majority were female (77.1%) and Caucasian (70.3%). Caregivers ranged in age from 41 to 65 years (M = 52.05, SD = 6.267). Almost half of the caregivers had a child with NF1 (48.6%), followed by a primary immunodeficiency (25.7%), cancer (20%), and then SCD (5.7%). Table 1 shows the demographic characteristics of the patient and caregiver samples.

### 3.2. Results from the Total Sample of Patients (All Medical Conditions)

Regarding the informational needs, the patients’ top three most endorsed items fell within the general health and wellness category. Specifically, these items included a need for more information about their primary medical condition (71.9%), nutrition and healthy eating (66.7%), as well as physical activity (66.7%) and pain management (66.7%). The three least endorsed items overall included information regarding attention problems (35.7%), keeping and making friends (33.3%), and spiritual and religious wellbeing and coping (23.6%). As far as service and program needs, just under two-thirds of patients endorsed a desire to connect with others who have a similar health condition via social media (63.10%), although there was much less of a desire for a structured online support group for AYAs with similar health conditions (38.1%). The least endorsed service need overall was teletherapy sessions at home with a therapist at our institution, although there was still a notable proportion of respondents who endorsed this need (33.3%). The item-by-item results of the patient survey are presented in Table 2, which includes the overall sample and a breakdown of responses for each medical condition.

### 3.3. Results from the Patient Sample: Between-Group Differences by Medical Condition

When comparing the responses between different medical conditions, the omnibus chi-squared tests indicated significant differences among the respondents’ interest in information about (a) nutrition and healthy eating (X^2^ (3) = 9.93, *p* < 0.05); (b) adjustment to diagnosis and treatment (X^2^ (3) = 8.37, *p* < 0.05); (c) adjustment to life post-treatment (X^2^ (3) = 13.09, *p*< 0.01); and (d) returning to school and/or work (X^2^ (3) = 23.46, *p*< 0.001). See Table 3 for results.

Post hoc partitioning of the data clarified where these differences occurred between the four medical conditions represented in this sample. For simplicity, only significant differences are reported herein. Regarding the desire for information about nutrition and healthy eating, individuals with SCD were significantly more likely to endorse this item than individuals with NF1 (X^2^ (1) = 4.90, *p* < 0.05) and primary immunodeficiency (X^2^ (1) = 7.36, *p* < 0.01). This same pattern was observed regarding the need for information about how to adjust to life post-treatment, as individuals with SCD were significantly more likely to endorse this need than individuals with NF1 (X^2^ (1) = 11.98, *p* < 0.02) and primary immunodeficiencies (X^2^ (1) = 5.97, *p* < 0.05). Individuals with SCD and cancer were much more likely to endorse a need for information about adjusting to their illness/treatment, compared to individuals with NF1 (X^2^ (1) = 4.52, *p* < 0.05 and X^2^ (1) = 5.76, *p* < 0.05, respectively). In addition, the need for more information about returning to school or work post-treatment was more commonly endorsed by individuals with SCD than those with NF1 (X^2^ (1) = 16.15, *p* < 0.001) and primary immunodeficiencies (X^2^ (1) = 8.67, *p* < 0.01). Individuals with cancer also were more likely to endorse a need for information about returning to school or work relative to those with NF1 (X^2^ (1) = 16.54, *p* < 0.001).

### 3.4. Results from the Patient Sample: Differences by Demographic Characteristics

#### 3.4.1. Patient Differences by Race

When comparing the responses to survey items among racial groups, several differences emerged. There were significant differences between respondents of different races on several items within the general health and wellness category. This section included the patients’ interest in learning more about (a) their medical condition (X^2^ (3) = 13.479, *p* < 0.01); (b) physical activity (X^2^ (3) = 11.32, *p* = 0.01); (c) interventions for sleep (X^2^ (3) = X^2^ (1) = 9.45, *p* < 0.05); and (d) treatment side effects (X^2^ (3) = 8.00, *p* < 0.05). Prior to discussing the post hoc comparisons, it is important to note that individuals with SCD represented 70% of Black respondents in this sample, reflecting a potentially meaningful confound. This finding should be taken into consideration when interpreting differences by race. See Table 4 for results.

Post hoc partitioning of significant findings indicated that Black respondents reported significantly more interest in learning about their medical condition compared to Caucasian (X^2^ (1) = 8.57, *p* < 0.005) and multiracial respondents (X^2^ (1) = 12.86, *p* < 0.001). In addition, compared to Caucasian participants, Black and multiracial respondents more commonly endorsed a desire for information about physical activity (X^2^ (1) = 9.08, *p* < 0.005 and X^2^ (1) = 4.25, *p* < 0.05, respectively) compared with all other races. Regarding an interest in learning about treatment side effects, Asian respondents endorsed this item more frequently than multiracial respondents (X^2^ (1) = 4.00, *p* < 0.05), while Black respondents had a greater endorsement rate relative to Caucasian respondents (X^2^ (1) = 4.06, *p* < 0.05). Finally, regarding interest in treatments for sleep difficulties, the only difference that emerged suggested greater endorsement by Asian respondents compared to Caucasian respondents (X^2^ (1) = 5.09, *p* < 0.05).

There were varying rates of endorsement among races on several other items throughout the needs survey. This included differences in the need for assistance/information regarding returning to school and/or work (X^2^ (3) = 14.50, *p* < 0.01), with Black and multiracial participants endorsing this item more frequently than Caucasian participants (X^2^ (1) = 11.32, *p* < 0.005 and X^2^ (1) = 6.70, *p* = 0.01, respectively). There was also a significant difference in respondents’ interest in teletherapy sessions (X^2^ (3) = 9.39, *p* < 0.05). Specifically, Black and Asian respondents had greater interest relative to the Caucasian respondents (X^2^ (1) = 8.73, *p* < 0.005 and X^2^ (1) = 4.10, *p* < 0.05, respectively). Furthermore, there was a significant difference in respondents’ interest in an AYA “hang out” space at the hospital (X^2^ (3) = 9.78, *p* < 0.05), with Black and multiracial respondents indicating a higher degree of interest (>70% endorsement), than Asian and Caucasian respondents (falling between 33–43%). Finally, there were differences among those needing help coping with uncertainty about the future (X^2^ (3) = 8.15, *p* < 0.05). Specifically, Black respondents endorsed this item more frequently than Caucasian respondents (X^2^ (1) = 5.12, *p* < 0.05).

#### 3.4.2. Patient Differences by Sex

There were several differences between the needs endorsed by male versus female respondents. Specifically, males endorsed significantly more interest in obtaining help to transition to an adult doctor (68.3%) compared to females (39.4%; X^2^ (1) = 6.18, *p* < 0.05). In contrast, females were more likely to endorse an interest in psychosocial therapeutic supports than males. This difference was significant for one-on-one counseling at the hospital (X^2^ (1) = 3.87, *p* < 0.05), teletherapy sessions while at home (X^2^ (1) = 5.17, *p* < 0.05), and web-based support groups for other AYAs with a similar health condition (X^2^ (1) = 6.00, *p* < 0.05). In addition, females more frequently endorsed a desire for an AYA “hang out” space at the hospital compared to males (X^2^ (1) = 4.33, *p* < 0.05). See Table 5 for results by gender.

#### 3.4.3. Patient Differences between Adolescent and Young Adult Respondents

Adolescent and young adult respondents were generally consistent in their endorsement of needs. The only significant difference pertained to the need for information about dating and love life (X^2^ (1) = 4.06 *p* < 0.05), with young adults endorsing this item more often than adolescents. Of note, these items had somewhat different wording, with the adolescent form referring to an interest in information about sexual identity and relationships and the young adult form referred to an interest in information about sexuality and intimacy. See Table 6 for results pertaining to age differences.

### 3.5. Caregivers’ Perspectives on Patient Needs

When caregivers were asked about their perception of their child’s needs, they most frequently endorsed interest in their child receiving more information about treatment side effects (70.3%). This was followed closely by a desire for information about their child’s primary medical condition (67.6%), nutrition/healthy eating (67.6%), impact of the diagnosis on their child’s school/work (67.6%), and transitioning from a pediatric to adult doctor (67.6%).

Among the 43 items included in both the caregiver and patient questionnaires, there was general consistency in their endorsements, suggesting that they have similar perceptions of the patients’ needs. Indeed, there were only three items for which caregivers and patients expressed significant differences. Specifically, caregivers less frequently endorsed that their child would have a need for information about health insurance (X^2^ (1) = 5.49, *p* < 0.05) or returning to school/work after treatment (X^2^ (1) = 8.91, *p* < 0.01). On the other hand, compared to the patients, caregivers more frequently endorsed that their child would benefit from web-based support groups (X^2^(1) = 6.06, *p* < 0.05). See Table 7 for results.

### 3.6. Open-Ended Perspectives of Patients and Caregivers

Patients and caregivers had the opportunity to provide open-ended comments at the end of the survey. Among the 19 AYA patients who provided comments, 13 individuals made comments about creating a social space in which they could pursue enjoyable, relaxing activities, and/or connect better with others who have health conditions. For example, one respondent shared that “(I would like) anything social, that’s not too overly intrusive… an opt-in social environment.” Another person explained, “Maybe have teen nights and games for young adults to meet and hang out.” Two AYAs reiterated the desire for more information about their health condition, and one person expressed a desire for better communication. Other comments were complimentary in nature.

Caregivers made several additional comments about enhancing social connections among AYAs with health conditions, and also among caregivers themselves. Out of 13 caregiver comments, 6 comments referenced a social space or opportunities for patients and caregivers to relax and connect with others. Four comments were complimentary in nature, and two regarded a need for more information about their child’s health condition and how they could best support their child’s wellbeing.

## 4. Discussion

The results of this needs assessment provide new insights into AYAs with chronic health conditions, including NF1, chronic or refractory cancer, primary immunodeficiencies, and SCD participating in clinical research. Overall, the survey suggested that there was an overwhelming need for general health and medical information from the respondents. Indeed, the most commonly endorsed item was the need for information about an individual’s primary health condition. This topic remained a significant concern when we examined the endorsement rates by disease groups, racial groups, and male and female sexes, which suggests that its importance cuts across medical and demographic variables.

While it may seem counterintuitive that AYA patients with longstanding health conditions would need more information about their disease, this particular finding is consistent with the extant literature and has been reported. For example, a systematic literature review of adolescent oncology patients suggested there is an unmet need for more information about a range of illness-related issues throughout the disease trajectory, including the illness itself, available medical therapies, and long-term effects of treatment [28]. This need is echoed in the SCD literature, in which adolescent patients with SCD reported a desire for self-management tools that included disease-based education [29].

Notably, 90% of Black respondents expressed a particular interest/need for information about the health condition, which was significantly greater than other racial groups surveyed. This finding is important because the U.S. healthcare system has an unfortunate history of disenfranchising Black and Brown patients [30]. As a healthcare community, it is important to recognize this historical context and make a concerted effort to equip minority patients with the knowledge and information to heed medical recommendations and promote wellbeing. More generally speaking, it is important to maintain awareness that AYA patients with different ethnic and racial background may maintain different healthcare needs. This pattern was observed in the current study and is important to consider when working with patients in different hospital and community settings.

Alongside the need for more information about an individual’s health condition, the results suggest a strong need for more information about physical activity and nutrition. These needs are nonspecific to having a chronic health condition, and are common issues raised in the context of primary care more generally. However, it is possible that individuals with rare and chronic health conditions are less likely to receive this information during their visits, because there are numerous other items to address. Further, despite the advantages of multidisciplinary care and efforts to promote a more integrated care system, symptom management has the potential to become siloed when so many providers are involved [31]. This may create a gap in the information communicated about overall wellbeing.

It is important that AYA patients feel they have a solid understanding of their specific health condition and healthy behavior in general. In fact, research suggests that an awareness of physical symptoms, disease risks, and health-promotion behaviors can help patients self-monitor their health and promote wellbeing [32]. In addition, health knowledge underlies compliance with treatments and recommendations made by medical providers [33,34]. Despite the well-known benefits of this information, interventions that promote disease knowledge and physical wellness are not universally integrated into AYA care. Addressing the need for disease knowledge entails relatively simple and cost-effective efforts, including nurse or physician directed education during appointments. It is, however, notable that information must be developmentally appropriate, and easily comprehendible [35]. In other words, information needs to be digestible in order to have an impact.

### 4.1. Least-Endorsed Items

Surprisingly, the results of the current survey suggested less desire for mental health or psychosocial services by patients. There may be several explanations that account for this finding. In particular, the hospital at which this survey was administered has a national and international catchment of patients. Respondents were asked if they would like to receive mental health services through our institution (in-person or virtually), and it is possible that these services are better accessed through their local hospital or communities. In fact, we do not know how many respondents already were receiving therapeutic services in their local area. Another possibility is that respondents are simply less interested in individual or family therapy as a traditional service. This result begs the question as to whether behavioral health clinicians should broaden how they think about best serving the chronically ill AYA community. Perhaps, some patients may be in greater need of assistance implementing health behaviors rather than addressing acute mental health conditions.

### 4.2. Differences by Sex

While there was generally modest interest in psychosocial supports (as explained above), it is notable that more females than males endorsed interest in these types of services. This is consistent with rates of engagement with the mental-health field more generally, in which females are more likely to seek behavioral health support compared to males [36]. In contrast, males expressed a greater need for support as they seek out an adult doctor. Historically, women tend to have higher healthcare utilization than men, so it is possible this finding reflects greater comfort by women in obtaining care or identifying a doctor [37].

### 4.3. Differences between Caregiver and Patient Respondents

The results of this study indicate strong convergence between caregiver and patient perceptions of need, suggesting that caregivers of individuals with chronic health conditions likely have a solid sense of what their child is seeking. This was further evidenced by convergence in open-ended comments made by caregivers and patients on the survey. Thus, providers would benefit from seeking parental input when trying to assess AYA patient needs. However, there were a few exceptions to these findings, such as parents (compared to AYAs) reporting less need for information about insurance and support to return to school/work, and a greater need for internet support groups for their child.

### 4.4. Limitations

The results of the current study should be viewed in the context of several limitations. In particular, we were limited by our sample in multiple ways. While the overall sample size was reasonable, it was far more limited when we parsed the sample into smaller groups, such as by racial group, age, or health condition. Comparisons between these subgroups were offered, but they have limited generalizability. In addition, the current sample included limited health conditions, including cancer, NF1, SCD, and a primary immunodeficiency. Thus, results cannot speak to the needs of AYAs with other chronic conditions. Further, the current survey was anonymized, which constrained additional analyses that would deepen our understanding of how patient characteristics relate to specific needs. This includes associations between how a patient’s point in the disease trajectory, or treatments, for example, may relate to their needs. These factors should be explored by future research. Finally, this study’s findings are limited by the somewhat unique setting in which the information was obtained. Specifically, our institution is typically a secondary site of medical care, as visiting patients have a provider or hospital nearer to their home community. Furthermore, all patients at our institution are enrolled on a clinical research protocol and many are undergoing experimental treatments. Thus, this patient population may reflect a slightly different set of needs than would be observed in a primary healthcare setting. It also bears noting that the current survey was administered prior to the COVID-19 pandemic. Therefore, the needs expressed by respondents in the current survey may be different, had the survey been administered following the onset of the pandemic.

### 4.5. Future Directions and Conclusions

There are several future directions based on this research. First, it would be beneficial for additional data to be collected pertaining to needs of AYAs with chronic health conditions across healthcare settings to compare the differences between the findings. While the current QI survey was developed to improve services at our institution, it can be easily administered and adapted across settings, which is encouraged. Future research and clinical initiatives also should focus on developing services that can better meet the needs of AYA patients. A particularly feasible target is disseminating information during routine visits. Such information can be delivered through verbal education, written materials, or a combination of approaches, and should be available virtually and in a hard-copy format. Moreover, research that utilizes digital technologies to expand services and address informational needs is particularly compelling, as 91% of AYAs have access to smartphone apps and web-based services [38]. Ultimately, with a more thorough assessment of needs and with the development of supports and services, we can make significant strides in improving the quality of life for vulnerable AYA patients with chronic health conditions.

## Figures and Tables

**Table 1 children-09-00500-t001:** Sociodemographic characteristics of survey respondents.

Variable	Patients M ± SD	Patients *n* (%)	Caregivers M ± SD	Caregivers *n* (%)
Age (years)	23.5 ± 5.9		52.1 ± 6.3	
Gender				
Male		53 (59.6)		8 (21.6)
Female		35 (39.3)		27 (73.0)
Not disclosed		1 (1.1)		2 (5.4)
Race				
White		37 (41.6)		26 (70.3)
Black		27 (30.3)		3 (8.1)
Asian		9 (10.1)		5 (13.5
American Indian or Alaska Native		3 (3.4)		1 (2.7)
Multiracial		8 (9.0)		1 (2.7)
Not disclosed		5 (5.6)		1 (2.7)

**Table 2 children-09-00500-t002:** A: Informational needs endorsed by AYAs with chronic health conditions. B: Service-related needs endorsed by AYAs with chronic health conditions.

Item	Cancer		NF1		SCD		Autoimmune		Total	
	*n*	%	*n*	%	*n*	%	*n*	%	*n*	%
My health condition	12	63.2	22	71.0	17	85.0	9	64.3	60	71.4
Nutrition/healthy eating	14	73.7	18	58.1	18	90.0	6	42.9	56	66.7
Physical activity and exercise	13	68.4	17	54.8	17	85.0	9	64.3	56	66.7
Pain	12	63.2	19	61.3	18	90.0	7	50.0	56	66.7
Treatment side effects	9	47.4	21	67.7	15	75.0	8	57.1	53	63.1
Stress management	12	63.2	18	58.1	13	65.0	9	64.3	52	61.9
Complementary healthcare	11	57.9	16	51.6	16	80.0	8	57.1	51	60.7
Impact of condition on school or work	13	68.4	13	41.9	15	75.0	10	71.4	51	60.7
Financial assistance	10	66.7	12	48.0	13	65.0	8	66.7	43	59.7
Medical insurance	11	57.9	19	61.3	13	65.0	7	50.0	50	59.5
Sleep difficulties	9	47.4	15	48.4	16	80.0	9	64.3	49	58.3
Eating and/or weight	13	68.4	15	48.4	13	65.0	6	42.9	47	56.0
Uncertainty about the future	12	63.2	13	41.9	14	70.0	8	57.1	47	56.0
Returning to school/work	14	73.7	8	25.8	18	90.0	7	50.0	47	56.0
Transitioning to adult care	11	57.9	17	54.8	11	55.0	8	57.1	47	56.0
Academic support in college	10	66.7	13	52.0	13	65.0	3	25.0	39	54.2
Fertility	12	63.2	14	45.2	12	60.0	7	50.0	45	53.6
Coping with condition/tx	9	47.4	15	48.4	14	70.0	7	50.0	45	53.6
Physical limitations	9	47.4	15	48.4	13	65.0	7	50.0	44	52.4
Jobs/careers	9	47.4	16	51.6	14	70.0	5	35.7	44	52.4
Adjust to life post-treatment	11	57.9	9	29.0	16	80.0	7	50.0	43	51.2
Anxiety/depression	9	47.4	16	51.6	10	50.0	7	50.0	42	50.0
Legal and practical services	12	63.2	12	38.7	10	50.0	6	42.9	40	47.6
Appearance	8	42.1	13	41.9	9	45.0	8	57.1	38	45.2
Dating, sexuality	9	47.4	13	41.9	11	55.0	5	35.7	38	45.2
Communicating w/med team	8	42.1	13	41.9	10	50.0	7	50.0	38	45.2
Difficult conversations	11	57.9	11	35.5	10	50.0	6	42.9	38	45.2
Adjusting to dx/tx	12	63.2	8	25.8	11	55.0	5	35.7	36	42.9
Discussing medical condition	9	47.4	11	35.5	7	35.0	5	35.7	32	38.1
Attaining social support	7	36.8	11	35.5	7	35.0	6	42.9	31	36.9
Learning difficulties	6	31.6	13	41.9	8	40.0	4	28.6	31	36.9
Attention problems	6	31.6	12	38.7	7	35.0	5	35.7	30	35.7
Keeping and making friends	7	36.8	11	35.5	5	25.0	5	35.7	28	33.3
Transitioning to college	2	50.0	1	16.7	0	0.0	1	50.0	4	33.3
Spiritual/religious coping	2	13.3	10	40.0	3	15.0	2	16.7	17	23.6
Connecting with others with a similar health condition on social media	13	68.4	19	61.3	15	75.0	6	42.9	53	63.1
A hang-out space for AYAs	11	57.9	14	45.2	15	75.0	8	57.1	48	57.1
An NIH event where experts would address AYA issues related to condition	11	57.9	16	51.6	13	65.0	6	42.9	46	54.8
Scheduled daytime recreational activities for AYAs	12	63.2	11	35.5	11	55.0	9	64.3	43	51.2
Website with information related to my health condition	10	52.6	13	41.9	13	65.0	6	42.9	42	50.0
Recreational weekend programs with other AYAs with my condition	11	57.9	10	32.3	13	65.0	8	57.1	42	50.0
Willing to make an extra visit to NIH for patient events	7	36.8	16	51.6	10	50.0	7	50.0	40	47.6
Electronic newsletter	9	47.4	13	41.9	12	60.0	5	35.7	39	46.4
Supportive counseling for patient at NIH	11	57.9	11	35.5	10	50.0	4	28.6	36	42.9
In person support group for AYAs with my condition	9	47.4	11	35.5	9	45.0	7	50.0	36	42.9
Supportive counseling for my family at NIH	8	42.1	10	32.3	9	45.0	5	35.7	32	38.1
Online support group for AYAs with my condition	10	52.6	9	29.0	9	45.0	4	28.6	32	38.1
Video chat sessions at home with therapist between visits	5	26.3	9	29.0	11	55.0	3	21.4	28	33.3

Note: the table lists informational needs first, ordered from the most to the least endorsed item, followed by service-related needs, ordered from most to the least endorsed item; abbreviations: tx = treatment; dx = diagnosis; SCD = sickle cell disease; NF1 = neurofibromatosis type 1; AYAs = adolescent and young adults; NIH = national institutes of health.

**Table 3 children-09-00500-t003:** Significant differences in endorsements among medical condition.

	Percent of Respondents Who Responded “Yes”				
	Cancer (%)	NF1 (%)	SCD (%)	PI (%)	X^2^ Statistic
Information re: returning to school/work	73.7	25.8	90	50	23.5 **
Information re: adjusting to life post-treatment	57.9	29.0	80	50	13.1 **
Information re: adjusting to diagnosis and treatment	63.2	25.8	55	35.7	8.4 *
Information re: nutrition and healthy eating	73.7	58.1	90	42.9	9.9 *

* *p* ≤ 0.05; ** *p* ≤ 0.01. Abbreviations: NF1 = neurofibromatosis type 1; SCD = Sickle cell disease; PI = Primary immunodeficiency.

**Table 4 children-09-00500-t004:** Significant differences in endorsements among racial groups.

	Percent of Respondents Who Responded “Yes”				
	White (%)	Black (%)	Asian (%)	Multirace (%)	X^2^ Statistic
Information re: primary medical condition	64.9	92.6	77.8	37.5	15.2 **
Information re: physical activity	46.9	81.5	88.9	87.5	15.1 **
Information about treatment for sleep	43.2	70.4	88.9	75	12.6 *
Information re: treatment side effects	56.8	77.8	88.9	37.5	13.8 *

* *p* ≤ 0.05; ** *p* ≤ 0.01.

**Table 5 children-09-00500-t005:** Significant differences in endorsements by gender.

	Percent of Respondents Who Responded “Yes”		
	Males (%)	Females (%)	X^2^ Statistic
Information re: transitioning to an adult provider	64.2	42.9	3.9 *
One-on-one counseling at hospital	35.9	57.1	3.9 *
Teletherapy sessions	22.6	45.7	5.2 *
Web-based support group	28.3	54.3	6.0 *
Interest in an AYA hang-out space	49.1	71.4	4.3 *

* *p* ≤ 0.05.

**Table 6 children-09-00500-t006:** Significant differences in endorsements by age.

	Percent of Respondents Who Responded “Yes”		
	Adolescents (%)	Young Adults (%)	X^2^ Statistic
Transitioning to an adult provider	21.4	50.1	4.1 *

* *p* ≤ 0.05.

**Table 7 children-09-00500-t007:** Significant differences in endorsements between patients and caregivers.

	Percent of Respondents Who Responded “Yes”		
	Patient (%)	Caregiver (%)	X^2^ Statistic
Information re: returning to school/work post-treatment	56.2	27.0	8.9 **
Information about health insurance	60.7	37.8	5.5 *
Interest in web-based support groups	38.2	62.2	6.1 *

* *p* ≤ 0.05, ***p* ≤ 0.01.

## Data Availability

The data that support the findings of this study are available from the corresponding author upon reasonable request.

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
