# Peer review of "The Needs of Adolescents and Young Adults with Chronic Illness: Results of a Quality Improvement Survey"

_children, 2022, doi:10.3390/children9040500_

Round 1

Reviewer 1 Report

There are not many studies in the literature that have reviewed this topic so this is an important contribution to the literature.  I would like to  know more about where the patients  are in their disease trajectory. For instance, how far from diagnosis are they, are they on therapy, long term follow up, on a phase I trial etc. AND how does this relate to the findings.  

Author Response

Response 1: Thank you very much for this comment. We completely agree with the importance of this level of analysis. Unfortunately, the anonymized nature of our data prevented us from collecting this information and exploring it fully. We did add these particular limitations to our discussion, and noted the importance of studying them in the future (lines 405-407).

Reviewer 2 Report

I think that the topic of this manuscript and approach are very interesting and reasonable and within the scope of the Journal.

I think there are only few concern tha Authors could better adress:

  • Check percentages in table 1
  • Insert a  table of chi quadro analysis to explain results
  • Report few examples about point of view of caregiver and patients (in available)

Author Response

Comment 1: Check percentages in table 1

Response 1: Thank you for noting the percentages! We had forgotten to include numbers of patients/caregivers who did not disclose their gender. Those numbers are now in the table and all percentage values were double checked.

Comment 2: Insert a  table of chi quadro analysis to explain results

Response 2: We appreciate the challenges of sifting through these results. Per your suggestion, we added tables that contain the results of the omnibus Chi Square tests in an effort to make the results easier to review for the reader. By providing the percentage of positive endorsements in a table, we feel it is simpler to perceive where differences between groups lie, which can be cross-referenced with the text in the manuscript.

Comment 3: Report few examples about point of view of caregiver and patients (in available)

Response 3: In an effort to provide more depth about the perspectives of caregivers and patients, we included a section in our results regarding open-ended commentary provided by respondents at the end of the survey. We hope this helps give further detail regarding the point of view of patients and caregivers. Unfortunately, because surveys were anonymized, we could not explicitly look at convergence among caregiver-patient dyads. We only have the ability to look at this more generally in our sample across all patient respondents compared to all caregiver respondents.